# Axial compression behavior and failure mechanism of CFRP-confined circular hollow steel tube short columns: Theoretical and numerical analysis

**Jian Chen[1], Hairong Huang[1]\*, Yun Zhou[1,2], Kan Liu[3]**

**1** Architectural Engineering Institute, Zhejiang Tongji Vocational College of Science and Technology, Hangzhou, Zhejiang, China, **2** College of Water Conservancy and Architectural Engineering, Shihezi University, Shihezi, Xinjiang, China, **3** College of Architecture and Environment, Sichuan University, Chengdu, Sichuan, China

\* z20120100803@zjtongji.edu.cn

## Abstract

Circular hollow steel tubes (CHST) are widely employed as short columns in various infrastructure applications. This study comprehensively investigates the mechanical behavior of CFRP-Confined CHST (CFRP-CHST) short columns under axial compression through theoretical research, finite element analysis, and existing experimental data. New theoretical formulas for calculating the yield and ultimate bearing capacities of CFRP-CHST short columns under axial loading are developed based on continuum mechanics and the limit equilibrium method. The accuracy and reliability of these formulas are validated through comparisons with finite element simulations and experimental results. Theoretical analysis reveals that CFRP provides only a modest enhancement to the yield-bearing capacity of CHST short columns. However, within a certain range of CFRP layers, the ultimate bearing capacity is significantly improved, albeit with limitations. The concept of the CFRP confinement coefficient is introduced to define the effective range in which CFRP reinforcement substantially enhances the bearing capacity of CHST short columns. The mechanisms restricting this enhancement are investigated in detail through experimental data and finite element analysis. This research offers valuable design and analysis methods for the engineering applications of CFRP-CHST short columns.

## 1. Introduction

A significant number of infrastructure projects feature hollow steel tube components, such as those in sports arenas, high-speed rail station canopies, offshore platforms, airport terminals, and wind turbine structures. The primary cross-sectional forms of these components include rectangular hollow steel tubes (RHST) [1,2], square hollow steel tubes (SHST) [3,4], and circular hollow steel tubes (CHST) [5,6]. Among these, CHST is widely used in civil engineering due to its uniform stiffness in all directions, efficient material utilization, and attractive appearance [5,6]. During operation, some steel tube components suffer from corrosion, while others weaken due to overloading or experience reduced strength from seismic damage or other factors, leading to the development of reinforcement technologies.

**Data availability statement:** All relevant data are within the paper and its Supporting Information files.

**Funding:** This work was supported by the General Research Project of the Zhejiang Provincial Department of Education (No. Y202352472). The funders had no role in study design, data collection and analysis, decision to publish, or preparation of the manuscript.

**Competing interests:** On behalf of all authors, disclose any competing interests that could be perceived to bias this work—acknowledging all financial support and any other relevant financial or non-financial competing interests.

In the past decade, carbon fiber reinforced polymer (CFRP) has gained widespread use in engineering structures due to its remarkable mechanical properties, including low weight and high strength. This is particularly evident in the strengthening of concrete columns and concrete-filled steel tube (CFST) columns [7,8]. Extensive experimental, theoretical, and numerical research has confirmed that CFRP reinforcement significantly improves the strength and ductility of these structures [9,10]. However, studies on CFRP reinforcement in metal structures are still limited. Most research focuses on CFRP reinforcement of hollow steel tubes under bending conditions [11–13], with fewer addressing the reinforcement of compressed components, particularly short columns. Theoretical research in this area is scarce, with most emphasis placed on experimental and numerical analyses.

Research on the reinforcement of CHST columns remains relatively limited. Ghanbari et al. [14] conducted experimental studies to examine the structural behavior of CFRP-confined composite columns under pure axial compression. They evaluated plastic buckling, failure modes, deformation responses, and the impact of various materials on the ultimate load-bearing capacity. Liu et al. [15] explored the mechanical properties of CFRP composite components under axial compressive loads through both experimental and numerical simulations, focusing on the influence of fiber winding orientation on buckling failure modes. Wei et al. [16] studied the failure process of CFRP-strengthened CHS short columns under unloading and preloading conditions, investigating the effects of steel tube wall thickness, diameter-to-thickness ratio, and CFRP wrapping method on reinforcement efficiency. Zu et al. [17] examined the mechanical properties of CFRP-confined hollow steel tube short columns under axial tensile loading, finding that thick-walled steel tubes effectively suppressed premature debonding failure of CFRP, improving reinforcement efficiency. Kumar and Senthil [18] investigated the mechanical performance of CFRP-CHST short columns under static and cyclic axial loads, concluding that CFRP reinforcement enhanced both strength and ductility. Haedir and Zhao [19] proposed a design method for CFRP-reinforced CHST short columns and conducted axial compression tests, demonstrating that CFRP reinforcement increased axial load capacity compared to theoretical values from various codes. Bambach and Jama [20] performed axial compression tests on CFRP-reinforced thin-walled SHST columns, showing that CFRP wrapping delayed local buckling. Bambach and Elchalakani [21] studied the impact resistance of CFRP-confined SHS columns under axial impact, revealing significant improvements in impact performance. Silvestre et al. [22] conducted experimental research on the nonlinear mechanical behavior of CFRP-reinforced cold-formed channel columns, finding that CFRP reinforcement enhanced both axial load capacity and nonlinear behavior. Shaat and Fam [23] conducted experiments on CFRP-reinforced SHST columns, observing effective increases in axial load capacity for short SHS columns, and reduced lateral deflection for long SHS columns, with improvements depending on the initial defects of the specimens. Teng and Hu [24] examined the mechanical performance of GFRP-confined CHST short columns under axial compression, concluding that fiber reinforcement is a viable strengthening method for CHST short columns susceptible to buckling, although the increase in ultimate bearing capacity was limited.

In summary, CFRP reinforcement technology significantly improves the strength and ductility of structural components, making it a promising method for strengthening. While research on CFRP reinforcement of CFST columns is well-established, studies on CFRP reinforcement for metal structures remain limited. Despite the widespread use of axially compressed components in engineering, computational theories and key technologies for CFRP reinforcement of axially compressed steel columns are underdeveloped, and a comprehensive theoretical framework has yet to be established. Specifically, failure mechanisms remain insufficiently explored. To address these gaps, this study combines theoretical research and

numerical analysis to assess the effectiveness of CFRP reinforcement for CHST short columns under axial compression. New formulas for calculating the yield and ultimate bearing capacities of CFRP-CHST short columns are proposed and validated through comparisons with experimental and finite element analysis results. The theoretical investigation also examines the effective range of CFRP reinforcement for CHST short columns and provides a mechanistic analysis of the limitations on bearing capacity improvement.

## 2. Theoretical analysis

In practice, the primary mechanism by which CFRP enhances the overall bearing capacity of CFST or CHST short columns is through circumferential confinement of the reinforced areas. CFRP exhibits exceptionally high unidirectional tensile strength, and studies have demonstrated its effectiveness in reinforcing CHST short columns that fail due to local buckling [19,24]. However, its effectiveness is limited in reinforcing short columns subjected to inward buckling deformation away from the column ends. The theoretical analysis is based on the limit equilibrium method, which does not require consideration of intermediate forces and deformation processes. Instead, it focuses on establishing force equilibrium equations at specific states. The yield-bearing capacity corresponds to the yielding of the steel tube section, while the ultimate bearing capacity is determined by the rupture of CFRP at localized bulging areas. A depiction of CFRP reinforcement of CHST components is shown in Fig 1.

The bearing capacity of CFRP-confined CHST short columns consists of two components: one provided by the steel tube itself and the other representing the increase in capacity due to CFRP reinforcement. Based on prior experimental results [19,24], several reasonable computational assumptions are proposed as follows:

(1) CFRP fabric is treated as a linearly elastic material, with tensile strength only in the fiber direction, while other mechanical properties are neglected.

(2) The bond between the steel tube and CFRP is assumed to be fully intact, with no separation occurring during loading, ensuring collaborative deformation of both materials [19,24].

(3) Steel is considered an isotropic material. For simplification in the bearing capacity calculation, the steel tube is assumed to behave as an ideal elastic-plastic material, following the Von Mises yield criterion.

Under axial compression, the primary load is carried by the steel tube of CFRP-CHST short columns, which undergoes circumferential expansion. The circumferential confinement

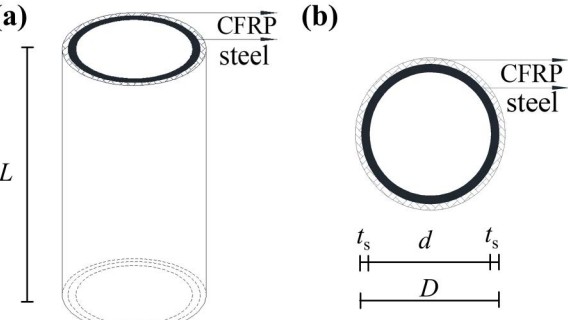

**Fig 1. CFRP-CHST short column: (a) elevation, and (b) cross-section.**

provided by the CFRP fibers limits the development of circumferential strain, resulting in a triaxial stress state for the steel tube. This stress state includes axial compression, circumferential compression, and radial compression of the outer wall. The circumferential confinement force provided by CFRP is denoted as $q_f$, with the thickness of the steel tube represented by $t_s$, the inner diameter by $d$, and the outer diameter by $D = d + 2t_s$. The yield strength is denoted as $f_y$. The force schematic for the steel tube and CFRP is illustrated in Fig 2.

When CFRP is applied with alternating longitudinal and circumferential layers, the equivalent cross-sectional method treats both types of wrapping as equivalent to circumferential wrapping. For the longitudinal wrapping method, only the contribution of the epoxy resin impregnated in the fabric is considered. The equivalent total thickness can be expressed as:

$$t_{eq,f} = \left( n_T + \frac{E_{L,f}}{E_{T,f}} n_L \right) t_f \tag{1}$$

where $E_{T,f}$ represents the tensile modulus in the principal fiber direction, and $E_{L,f}$ denotes the tensile modulus perpendicular to the principal fiber direction, which is taken as the modulus of the resin. $n_H$ represents the number of winding layers of the circumferential fibers, and $n_L$ represents the number of winding layers of the longitudinal fibers.

Assuming the steel tube behaves as an ideal elastic-plastic material, the Mises yield criterion provides the following relationship:

$$f_y = \sqrt{\frac{1}{2} \left[ \left( \sigma_{zs} - \sigma_{\theta s} \right)^2 + \left( \sigma_{zs} - \sigma_{rs} \right)^2 + \left( \sigma_{\theta s} - \sigma_{rs} \right)^2 \right]} \tag{2}$$

where $f_y$ represents the yield strength of steel, and $\sigma_{zs}$, $\sigma_{rs}$, and $\sigma_{,s}$ represent the axial stress, radial stress, and circumferential stress of the steel tube, respectively.

Establish the static equilibrium conditions based on Fig 2a.

$$\begin{cases} \int_{d/2}^{d/2+t_s} \sigma_{,s} d\rho = \dfrac{q_f d}{2} \\ \int_{d/2+t_s}^{d/2+t_s+t_{eq,f}} \sigma_{,f} d\rho = \dfrac{-q_f \left( d + 2t_s \right)}{2} \end{cases} \tag{3}$$

where $\sigma_{,f}$ represents the circumferential stress of CFRP, $q_f$ represents the circumferential confinement force provided by CFRP, and $d$ and $t_s$ represent the inner diameter and wall thickness of the steel tube, respectively.

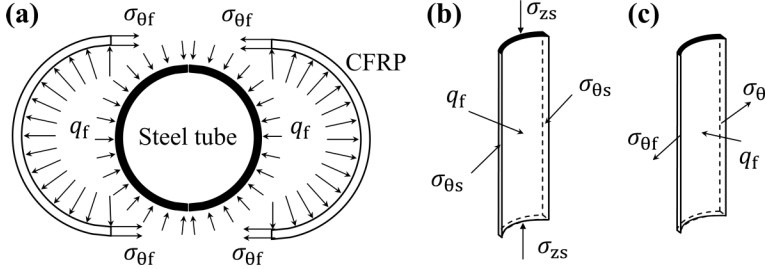

**Fig 2. Stress analysis of CFRP-CHST short columns: (a) overall stress distribution, (b) stress state of the reinforced steel tube, and (c) stress state of CFRP.**

The axial stress in the steel tube under CFRP confinement is obtained from Eq. (2) as

$$\sigma_{zs} = \sqrt{f_y^2 - \frac{3}{4}\left(\sigma_{\theta s} - \sigma_{rs}\right)^2} + \frac{\sigma_{\theta s} + \sigma_{rs}}{2} \tag{4}$$

The bearing capacity of CFRP-reinforced CHST short columns can be expressed as

$$N = \sigma_{zs} A_s \tag{5}$$

In Eq. (5), the axial stress of the steel tube is influenced by the stress-strain behavior of CFRP. The confining effect of CFRP enhances the buckling resistance and stability of the steel tube, thereby improving the bearing capacity of the component.

## 2.1. Yield bearing capacity

For an ideal elastic-plastic steel tube that adheres to generalized Hooke's law before yielding, the circumferential strain is given by

$$\varepsilon_{\theta s} = \frac{1}{E_s}\left[\sigma_{\theta s} - \mu_s\left(\sigma_{zs} + \sigma_{rs}\right)\right] \tag{6}$$

where $E_s$ and $\mu_s$ represent the elastic modulus and Poisson's ratio of the steel material, respectively.

Unidirectional tensile fiber CFRP, as a linearly elastic material, follows Hooke's law before fracture. The circumferential strain is given by

$$\varepsilon_{\theta f} = \frac{\sigma_{\theta f}}{E_{T,f}} \tag{7}$$

where $\varepsilon_{\theta f}$, $\sigma_{\theta f}$, and $E_{T,f}$ represent the circumferential strain, stress, and modulus of elasticity of CFRP, respectively.

Based on the assumption (2) and the deformation compatibility condition ($\varepsilon_{\theta f} = \varepsilon_{\theta s}$), the circumferential stress in the steel tube ($q_{f,y}$) due to CFRP at the overall yielding of the CHST short column can be determined by simultaneously solving Eqs. (3), (4), (6), and (7).

$$q_{f,y} = \psi f_y \tag{8}$$

where

$$\psi = \frac{\mu_s}{\sqrt{\left(\dfrac{E_s \gamma}{E_f} - \beta + \dfrac{\mu_s\left(\beta + 3\right)}{2}\right)^2 + \dfrac{3}{4}\mu_s^2\left(\beta - 1\right)^2}} \tag{9}$$

$$\begin{cases} \beta = \dfrac{d}{2t_s} \\ \gamma = \dfrac{D}{2t_{eq,f}} \end{cases} \tag{10}$$

On this basis, the three principal stress components at the yielding point of the steel tube can be determined. The radial stress is given by

$$\sigma_{rs} = \psi \cdot f_y \tag{11}$$

The circumferential stress is given by

$$\sigma_{\theta s} = \frac{\psi d}{2t_s} \cdot f_y \tag{12}$$

The axial stress is given by:

$$\sigma_{zs} = \left\{ \sqrt{1 - \frac{3}{4}\left[(\beta-1)\psi\right]^2} + \frac{(\beta+1)\psi}{2} \right\} \cdot f_y \tag{13}$$

Before yielding, the steel tube behaves as a linear elastic material, following Hooke's law. The longitudinal strain of the steel tube can be expressed as

$$\varepsilon_{zs} = \frac{1}{E_s}\left[\sigma_{zs} - \mu_s\left(\sigma_{\theta s} + \sigma_{rs}\right)\right] \tag{14}$$

By combining Eqs. (11) through (14), the longitudinal strain of the steel tube at the yielding point of the short column can be obtained as

$$\varepsilon_{zs} = \Gamma \frac{f_y}{E_s} \tag{15}$$

where

$$\Gamma = \sqrt{1 - \frac{3}{4}\left[(\beta-1)\psi\right]^2} + \left[\frac{(\beta+1)}{2} - (\beta+1)\mu_s\right]\psi \tag{16}$$

From condition $\varepsilon = \Delta L / L$, and by simultaneously solving Eqs. (15) and (16), the longitudinal displacement of the CHST short column at yielding can be obtained as

$$\Delta l_{sf,y} = \Gamma L \frac{f_y}{E_s} \tag{17}$$

where $L$ is the axial length of the CHST short column.

By combining Eqs. (5) and (13), the calculation formula for the yield-bearing capacity of CFRP-reinforced CHST short columns can be expressed as

$$N_{sf,y} = \left( \sqrt{1 - \frac{3}{4}\left[(\beta-1)\psi\right]^2} + \frac{(\beta+1)\psi}{2} \right) A_s f_y \tag{18}$$

In Eq. (18), the parameter $\psi$ is determined by the geometric and material properties of both the steel tube and CFRP. The confinement provided by CFRP enhances the yield-bearing capacity of the CFRP-CHST short column.

When the number of CFRP layers is zero, the formula reduces to the yield-bearing capacity calculation formula for CHST short columns,

$$N_{s,y} = A_s f_y \tag{19}$$

## 2.2. Ultimate bearing capacity

The failure mode of unreinforced CHST short columns, as observed in experiments from references [19,24], is illustrated in Fig 3a. Typically, the steel tube fails through localized outward bulging near the ends. After yielding, the deformation of the CHST short column increases sharply, causing a significant drop in load capacity and indicating poor ductility. In calculations, the yield limit of the steel tube is considered the strength limit of the short column.

For CFRP-reinforced CHST short columns with minimal confinement effects, the typical failure mode is shown in Fig 3b. This failure mode is primarily characterized by outward bulging at both ends of the column. Unlike unreinforced CHST short columns, which experience a rapid decrease in bearing capacity due to instability from outward bulging after yielding, CFRP-reinforced CHST short columns maintain their bearing capacity at this stage. The circumferential confinement provided by CFRP enhances the bearing capacity, improving ductility. As axial load increases, the CFRP in the bulging areas may eventually fracture due to excessive local deformation, reaching the ultimate tensile strength of the fibers. At this point, the bearing capacity peaks and then begins to decline, with the peak representing the ultimate bearing capacity of the component.

Based on the above analysis, for specimens with a low CFRP confinement coefficient, the ultimate bearing capacity of the CHST short column is determined by the fracture of the CFRP in the localized bulging area when it reaches the ultimate tensile strength. This corresponds to $\sigma_{\theta f} = f_f$. At this stage, the compressive stress exerted by the CFRP on the outer wall of the steel tube is given by

$$q_{f,l} = \frac{2 f_f t_{eq,f}}{d + 2 t_s} \tag{20}$$

Based on assumption (2), where the steel tube and CFRP deform collaboratively, the circumferential strain in the steel tube can be expressed as

$$\varepsilon_{\theta s} = f_f / E_{Tf} \tag{21}$$

By combining Eqs. (3)–(5) and Eq. (20), the calculation formula for the ultimate bearing capacity of CFRP-reinforced CHST short columns can be derived as

$$N_{sf,l} = \left( \sqrt{f_y^2 - 3 \left[ \frac{(\beta - 1) f_f t_{eq,f}}{D} \right]^2} + \frac{(\beta + 1) f_f t_{eq,f}}{D} \right) A_s \tag{22}$$

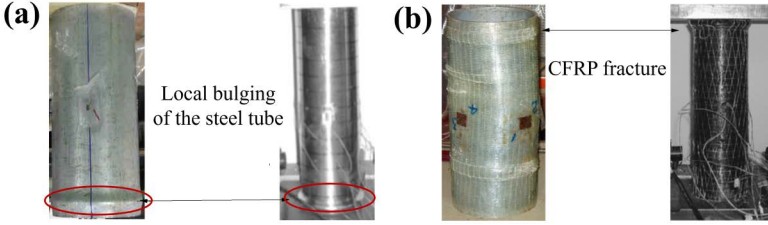

**(a)** Local bulging of the steel tube **(b)** CFRP fracture

**Fig 3. Typical failure modes of the specimens in experiments [19,24]: (a) CHST short columns, and (b) CFRP-CHST short columns.**

where $A_s = \pi D t_s$, $D = d + 2t_s$, $\beta = \dfrac{d}{2t_s}$.

As shown in Eq. (22), the tensile strength of CFRP ($f_f$) and the reinforcement thickness ($t_{eq,f}$) have a significant impact on the ultimate bearing capacity of the CFRP-CHST short column.

In summary, the bearing capacity of CFRP-CHST short columns is categorized into yield-bearing capacity and ultimate bearing capacity, corresponding to the yield and ultimate states of the component, respectively. Eq. (18) can be used to calculate the yield-bearing capacity of the CFRP-CHST short column, while Eq. (22) can be used to calculate the ultimate bearing capacity of the CFRP-CHST short column.

## 2.3. Failure mechanism

It is important to note that, according to Eq. (22), the expression is meaningful under the condition that the circumferential confinement provided by the CFRP is sufficient to prevent premature failure of the steel tube, meaning that the CFRP must not fracture before the steel tube reaches its yield or ultimate capacity. This ensures that the CFRP can provide effective reinforcement and enhance the overall bearing capacity of the CHST short column.

$$f_y^2 - 3\left[\frac{(\beta-1)f_f t_{eq,f}}{D}\right]^2 \geq 0 \tag{23}$$

That is, the parameters of the specimen should satisfy the following conditions:

$$\frac{\sqrt{3}(d-2t_s)f_f t_{eq,f}}{2Dt_s f_y} \leq 1 \tag{24}$$

To facilitate practical engineering applications, the above conditions are simplified. In practical engineering, CHST short columns typically satisfy the condition $d \gg 2t_s$. Under this assumption, $D \approx d$, and the expression can be simplified to

$$\frac{f_f t_{eq,f}}{f_y t_s} \leq \frac{2\sqrt{3}}{3} \tag{25}$$

Introducing the CFRP confinement coefficient for the steel tube

$$\xi_f = \frac{A_{cfrp} f_f}{A_s f_y} \tag{26}$$

Substituting the cross-sectional areas $A_{cfrp} \approx \pi D t_{eq,f}$ for the CFRP and $A_s \approx \pi D t_s$ for the steel tube, Eq. (26) can be further simplified to

$$\xi_f \leq \frac{2\sqrt{3}}{3} \approx 1.155 \tag{27}$$

That is, when reinforcing CHST short columns with CFRP in engineering practice, the CFRP confinement coefficient should not exceed 1.155. If this condition is not satisfied, the strength of the CFRP will not be fully utilized, and increasing the number of CFRP layers will not result in a significant improvement in the ultimate bearing capacity of the specimen. Consequently, the enhancement in the ultimate bearing capacity of CFRP-reinforced CHST short columns is limited.

## 3. Numerical analysis

A detailed finite element (FE) model was developed to analyze the mechanical performance of CFRP-reinforced CHST short columns and validate the theoretical derivations and their applicability by comparing them with experimental results. The analysis process primarily accounted for material nonlinearity, geometric nonlinearity, and contact nonlinearity. While the theoretical analysis assumes an idealized state, the experimental specimens may contain inherent material and geometric imperfections. To better replicate the actual conditions of the specimens, a buckling analysis was first performed to extract the first-order buckling mode of the model, which was then introduced as an initial imperfection in the load analysis phase.

Haedir and Zhao [19] conducted experiments on 10 CHST short columns, comprising four unreinforced specimens (C-1, C-2, C-3, and C-4) and six CFRP-reinforced specimens (CF-1A, CF-1B, CF-2A, CF-2B, CF-3A, and CF-4A). Each specimen had a length of 275 mm. The steel tube's measured elastic modulus was $2.10 \times 10^5$ MPa, with a yield strength of 455 MPa and a Poisson's ratio of 0.3. The tensile modulus of CFRP in the principal direction was $2.30 \times 10^5$ MPa, with a tensile strength of 1830 MPa, and the thickness of each CFRP layer was 0.176 mm. Additional specimen parameters are provided in Table 1. Teng and Hu [24] conducted experiments on four CHST short columns, including one unreinforced specimen (ST1) and three GFRP-reinforced specimens (ST2, ST3, ST4). The length of each specimen was 165 mm. The steel tube's measured elastic modulus was $2.01 \times 10^5$ MPa, with a yield strength of 455 MPa and a Poisson's ratio of 0.3. The tensile modulus of GFRP in the principal direction was $8.10 \times 10^4$ MPa, with a tensile strength of 1825.5 MPa, and the thickness of each GFRP layer was 0.17 mm. Parameters for the other specimens are listed in Table 1.

Following the theoretical analysis and simplifying the solution process, the constitutive relations for both the steel tube and the loading plates are modeled using an ideal elastic-plastic approach. This study focuses on the mechanical behavior and failure mechanisms of CFRP-CHST under axial compression. To achieve this, it is crucial to ensure that the model accurately represents the interaction between the steel column and CFRP, as well as their response to axial loading.

**Table 1. Model parameters.**

| Ref | Group | Specimen ID | $D$ /mm | $t_s$ /mm | $n_H$ | $n_L$ |
|---|---|---|---|---|---|---|
| [19] | 1 | C-1 | 87.25 | 2.36 | 0 | 0 |
| | | CF-1A | 87.23 | 2.32 | 1 | 1 |
| | | CF-1B | 87.21 | 2.32 | 2 | 2 |
| | 2 | C-2 | 86.31 | 2 | 0 | 0 |
| | | CF-2A | 86.38 | 1.96 | 1 | 1 |
| | | CF-2B | 86.39 | 1.96 | 2 | 2 |
| | 3 | C-3 | 85.75 | 1.57 | 0 | 0 |
| | | CF-3A | 85.68 | 1.57 | 1 | 1 |
| | 4 | C-4 | 85.11 | 1.13 | 0 | 0 |
| | | CF-4A | 85.21 | 1.1 | 1 | 1 |
| [24] | 5 | ST1 | 165 | 4.2 | 0 | 0 |
| | | ST2 | 166 | 4.2 | 1 | 0 |
| | | ST3 | 165 | 4.2 | 2 | 0 |
| | | ST4 | 165 | 4.2 | 3 | 0 |

Note: $D$ is the outer diameter of the steel tube, $t_s$ is the thickness of the steel tube, $n_H$ is the number of winding layers for circumferential fibres, and $n_L$ is the number of winding layers for longitudinal fibers.

In the experiments, the loading plates are rigid structures used to apply axial compressive forces. Given that the stiffness of the loading plates is significantly higher than that of the steel tube, it is reasonable to assume that the plates experience negligible deformation during the loading process. Additionally, the primary function of the loading plates is to transfer the load rather than to bear it. Therefore, their internal stress and deformation are not essential for understanding the mechanical behavior of CHST short columns. Based on these considerations, the loading plates at both ends were modeled as rigid bodies, simplifying the model and enhancing computational efficiency without sacrificing accuracy.

The CFRP is modeled using the Lamina model in Abaqus, which accurately captures the orthotropic anisotropy of CFRP. Since CFRP is alternately bonded in the longitudinal and circumferential directions in the experiments, it is necessary to define the material directions accordingly. For fibers wrapped in the circumferential direction, the material orientation is defined as shown in Fig 4a. The 1-direction corresponds to the principal fiber direction, with the elastic modulus set according to the tensile modulus of the principal fiber, denoted as $E_1$ in the model. The 2-direction is orthogonal to the principal fiber direction and has no fiber strength, represented as $E_2$ in the model. Under axial compression, shear failure of the material is not considered, and the shear modulus is set to 1. The CFRP elastic parameters of the Lamina model are defined according to the experimentally measured values. The Hashin failure criterion is used to model the progressive failure of CFRP during the loading process, with the tensile strength in the principal direction determined by the fiber strength. Extensive research has shown that CFRP exhibits high tensile resistance but limited compressive resistance. In this analysis, the compressive and shear strengths of CFRP are not considered and are both set to 1 MPa. The failure parameters for CFRP, as defined by the Hashin criterion, and the tensile strength of the fibers are based on experimentally measured data [19,24]. For fibers applied in the longitudinal direction, the remaining parameters are defined similarly to those for the circumferential direction, with the only modification being the orientation of the CFRP layup, as illustrated in Fig 4b.

Experimental results from the literature indicate that CFRP and the steel tube can deform collaboratively during loading, with no delamination occurring before failure [19,24]. In the finite element model, perfect bonding is assumed between the steel tube and CFRP, meaning no relative slippage occurs at the interface during the loading process. Binding constraints (tie) are applied between the steel tube and CFRP, as well as between the CFRP layers, to ensure the collaborative deformation of both materials.

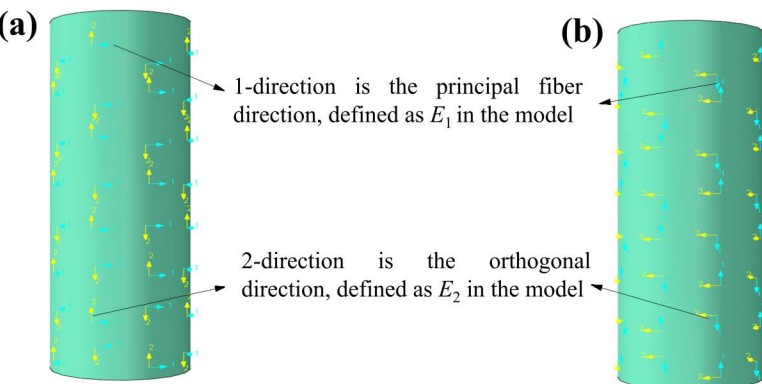

**Fig 4. CFRP material orientation definitions: (a) circumferential, and (b) longitudinal.**

To facilitate the extraction of the load-displacement curve, a reference point is established 20 mm from the loading end and is coupled with the upper surface of the loading end. Displacements are restricted at the lower end of the model, while all displacements except for the axial direction are constrained at the upper end. An axial displacement of 20 mm is then applied at the reference point. The steel tube is modeled using four-node reduced integration shell elements (S4R), while the CFRP is represented by four-node reduced integration membrane elements (M3D4R). Convergence testing has shown that setting the mesh size to 1/40 of the total model length optimizes computational efficiency while satisfying accuracy requirements.

## 4. Validation of theoretical and FE models

### 4.1. Load-displacement curves and failure modes

The load-displacement curve and failure modes obtained from the finite element (FE) model are compared with the experimental results of Haedir and Zhao [19], as shown in Fig 5. It is important to note that the failure modes for specimens CF-1A, CF-1B, and CF-3A are not reported in the literature; thus, the failure modes shown in the figure for these specimens are based solely on the FE model results. Due to factors such as instrument and operational errors during the experiments, perfect consistency between the FE model and experimental conditions is not achievable, leading to discrepancies within a reasonable range. Regarding curve trends, the overall stiffness of the load-displacement curve from the FE model is slightly higher than that of the experimental results. However, the general trend remains consistent with the experimental curve, and the ultimate load-bearing capacity predicted by the FE model closely aligns with the experimental results. The failure modes indicate that, under identical CFRP parameters, specimens with varying steel tube thicknesses exhibit different failure modes. Furthermore, for specimens with the same steel tube thickness, increasing the number of CFRP layers also alters the failure modes. This analysis demonstrates that the FE model developed in this study effectively simulates both the failure process and failure modes of the specimens.

### 4.2. Load-bearing capacity

To verify the accuracy of the proposed theoretical formula, the theoretical values are validated against the results obtained from the established FE model and the experiments [19,24].

**4.2.1. Yield load-bearing capacity.** Using Eqs. (15)–(17), the theoretical values for the longitudinal strain ($\varepsilon_{\mathrm{zs,y}}$) and longitudinal displacement ($\Delta l_{\mathrm{sf,y}}$) at the yield point of CFRP-CHST short columns can be calculated. The load-displacement curve obtained from the FE model enables the extraction of the longitudinal displacement ($\Delta l_{\mathrm{FE,y}}$) at the yield point of the CFRP-CHST short columns. The comparison results are presented in Table 2. The longitudinal displacement from the FE model is slightly higher than the theoretical solution. The average ratio between the theoretical and finite element values is 0.96, with a standard deviation of 0.0055. This suggests that the theoretical formula proposed in this study accurately predicts the axial deformations of the specimens.

From Table 2, the average values of the longitudinal strain and longitudinal displacement at the yield point of CFRP-CHST short columns are $\varepsilon_{\mathrm{FE,y}} \approx 2291\mu\varepsilon$ and $\Delta l_{\mathrm{FE,y}} \approx 0.63$ mm, respectively. In Fig 6, a reference line at $x \approx 0.63$ mm is drawn on the load-displacement curve obtained from the finite element analysis. It can be observed that the yield points from the finite element calculations are located near the reference line. The load at the intersection of the reference line and the curve corresponds to the FE value ($N_{\mathrm{FE,y}}$) of the yield load for the specimen.

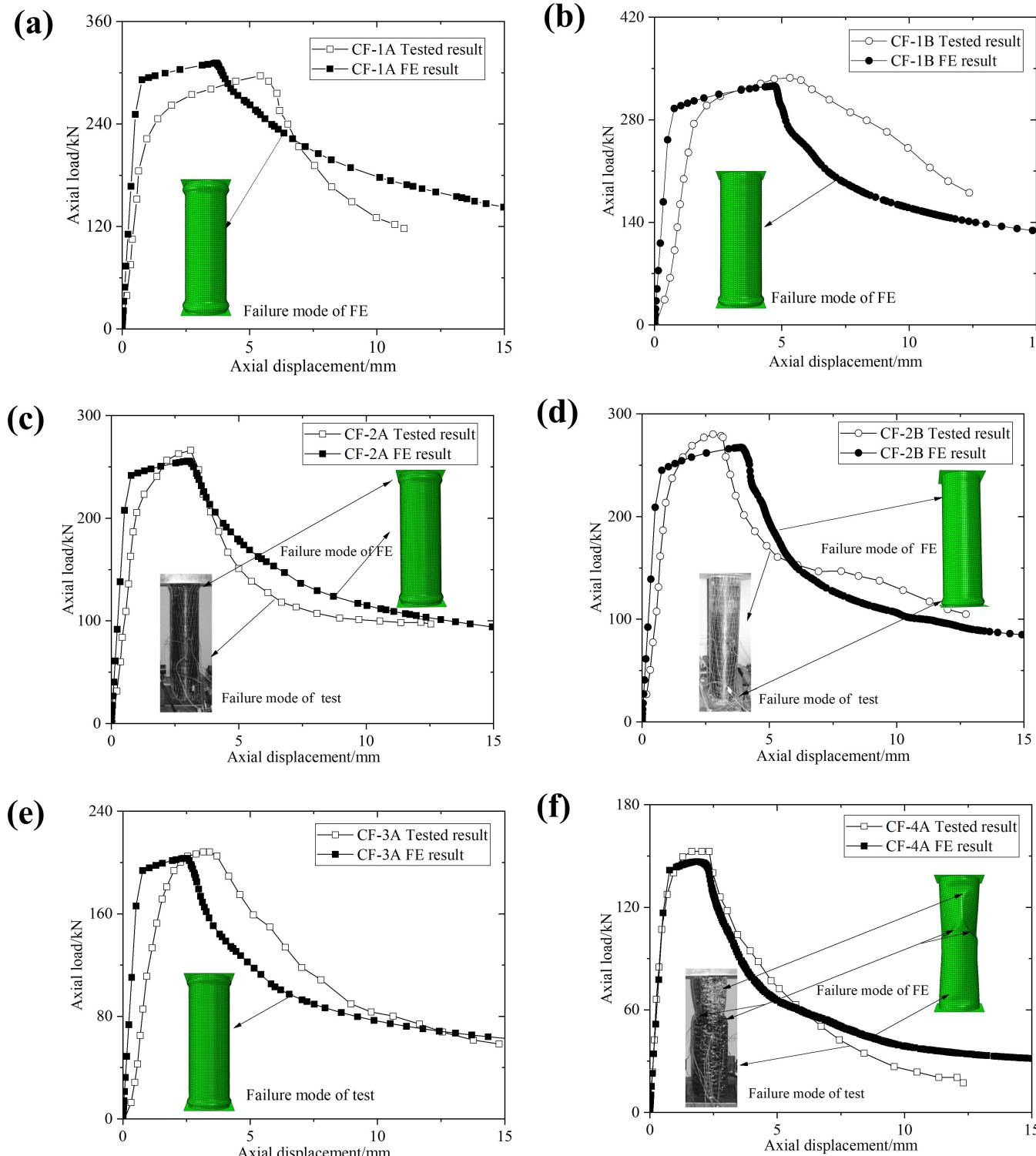

**Fig 5. Comparison of load-displacement curves and failure modes between FE model and experiment: (a) CF-1A, (b) CF-1B, (c) CF-2A, (d) CF-2B, (e) CF-3A, and (f) CF-4A.**

**Table 2. Comparison of longitudinal displacement at yield point for CHST short columns.**

| Ref | Group | Specimen ID | $L$ /mm | $\varepsilon_{zs,y}$ / $\mu\varepsilon$ | $\Delta l_{sf,y}$ /mm | $\Delta l_{FE,y}$ /mm | $\dfrac{\Delta l_{sf,y}}{\Delta l_{FE,y}}$ |
|---|---|---|---|---|---|---|---|
| [19] | 1 | C-1 | 275 | 2171 | 0.5971 | 0.6183 | 0.97 |
| | | CF-1A | 275 | 2183 | 0.6004 | 0.6217 | 0.97 |
| | | CF-1B | 275 | 2197 | 0.6042 | 0.6254 | 0.97 |
| | 2 | C-2 | 275 | 2171 | 0.5971 | 0.6267 | 0.95 |
| | | CF-2A | 275 | 2186 | 0.6010 | 0.6307 | 0.95 |
| | | CF-2B | 275 | 2202 | 0.6055 | 0.6352 | 0.95 |
| | 3 | C-3 | 275 | 2171 | 0.5971 | 0.6219 | 0.96 |
| | | CF-3A | 275 | 2189 | 0.6021 | 0.6269 | 0.96 |
| | 4 | C-4 | 275 | 2171 | 0.5971 | 0.6198 | 0.96 |
| | | CF-4A | 275 | 2197 | 0.6043 | 0.6270 | 0.96 |
| [24] | 5 | ST1 | 450 | 1660 | 0.7469 | 0.7765 | 0.96 |
| | | ST2 | 450 | 1661 | 0.7476 | 0.7765 | 0.96 |
| | | ST3 | 450 | 1663 | 0.7484 | 0.7766 | 0.96 |
| | | ST4 | 450 | 1665 | 0.7492 | 0.7766 | 0.96 |
| Average | | | | | | | 0.96 |
| Standard deviation | | | | | | | 0.0048 |

Note: $L$ is the length of the CFRP-CHST short column, $\varepsilon_{zs,y}$ is axial yield strain of the steel tube, $\Delta l_{sf,y}$ is the theoretical value of the axial displacement at the yield point of the CFRP-CHST short column, $\Delta l_{FE,y}$ is the FE value of the axial displacement at the yield point of the CFRP-CHST short column.

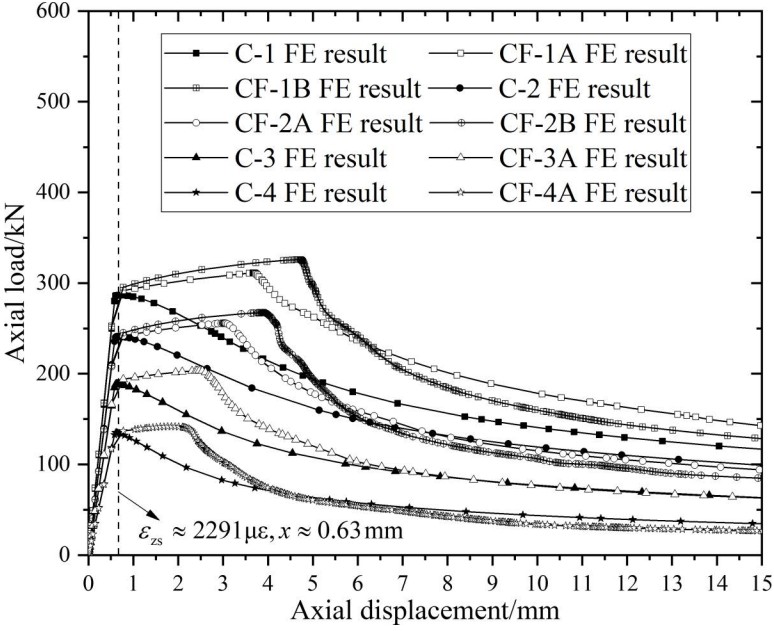

**Fig 6. FE results for the load-displacement curve of CFRP-CHST short columns.**

By combining Eqs. (18) and (19), the yield load $N_{s,y}$ of the unreinforced CHST short column and the yield load $N_{sf,y}$ of the CFRP-reinforced column can be calculated. Consequently, the enhancement ratio of the yield load due to CFRP reinforcement can be expressed as

$$W_{sf,y} = \frac{N_{sf,y} - N_{s,y}}{N_{s,y}} \times 100\% \qquad (28)$$

The FE values $N_{FE,y}$, theoretical values $N_{sf,y}$, and enhancement ratios $W_{sf,y}$ for the yield load of each specimen are presented in Table 3. For the yield load values, the average ratio of $N_{sf,y} / N_{FE,y}$ is 1.01, with a standard deviation of 0.0227. This indicates that the FE analysis results for the yield load are in good agreement with the theoretical calculations, validating the accuracy of the proposed theoretical formula. Regarding the enhancement ratio of the yield load, CFRP reinforcement does not result in significant improvement, with the maximum enhancement ratio being only 3.86% [19]. This limited enhancement is because, when the steel tube reaches its yield strength across the entire cross-section, the circumferential strain in the steel tube is relatively small, preventing CFRP from fully utilizing its strength. Further-more, with a relatively small number of CFRP layers, the calculated CFRP-induced circumfer-ential compressive stress $q_{f,y}$ on the steel tube (as shown in Table 3) is less than 2 MPa at most. Consequently, the CFRP provides minimal circumferential confinement to the CHST short columns, leading to only a modest increase in yield strength. This finding is consistent with the experimental results of Teng and Hu [24], as shown in Fig 7.

**4.2.2. Ultimate load-bearing capacity.** After the yield of CFRP-CHST short columns, as the load continues to increase, the circumferential strain of the CFRP gradually increases. This allows the CFRP's high tensile strength to be fully utilized, further enhancing the circumferential confinement it provides to the CHST short columns. The fundamental reason is that, according

**Table 3. Comparison of theoretical, experimental, and finite element values for yield and ultimate load-bearing capacity.**

| Ref | Group | Specimen ID | $t_s$ / mm | $N_{s,y}$ / kN | Yield load-bearing capacity | | | | | Ultimate load-bearing capacity | | | | | | | |
|---|---|---|---|---|---|---|---|---|---|---|---|---|---|---|---|---|---|
| | | | | | $q_{f,y}$ / Mpa | $N_{sf,y}$ / kN | $W_{sf,y}$ / % | $N_{FE,y}$ / kN | $\frac{N_{sf,y}}{N_{FE,y}}$ | $q_{f,l}$ / MPa | $N_{sf,l}$ / kN | $W_{sf,l}$ / % | $N_{FE,l}$ / kN | $N_{e,l}$ / kN | $\frac{N_{sf,l}}{N_{e,l}}$ | $\frac{N_{sf,l}}{N_{FE,l}}$ | $\frac{N_{FE,l}}{N_{e,l}}$ |
| [19] | 1 | C-1 | 2.36 | 286 | 0.0 | 286 | 0.00 | 280 | 1.02 | 0.0 | 286 | 0.00 | 280 | 253 | 1.13 | 1.02 | 1.11 |
| | | CF-1A | 2.32 | 281 | 0.7 | 286 | 1.45 | 285 | 1.00 | 7.4 | 317 | 12.51 | 312 | 299 | 1.06 | 1.01 | 1.04 |
| | | CF-1B | 2.32 | 281 | 1.6 | 290 | 3.18 | 287 | 1.01 | 14.9 | 334 | 18.70 | 330 | 341 | 0.98 | 1.01 | 0.97 |
| | 2 | C-2 | 2 | 241 | 0.0 | 241 | 0.00 | 235 | 1.03 | 0.0 | 241 | 0.00 | 235 | 220 | 1.10 | 1.03 | 1.07 |
| | | CF-2A | 1.96 | 236 | 0.7 | 240 | 1.73 | 240 | 1.00 | 7.5 | 270 | 14.01 | 261 | 267 | 1.01 | 1.03 | 0.98 |
| | | CF-2B | 1.96 | 236 | 1.7 | 246 | 3.86 | 243 | 1.01 | 15.0 | 280 | 18.34 | 273 | 281 | 1.00 | 1.02 | 0.97 |
| | 3 | C-3 | 1.57 | 189 | 0.0 | 189 | 0.00 | 183 | 1.03 | 0.0 | 189 | 0.00 | 183 | 170 | 1.11 | 1.03 | 1.08 |
| | | CF-3A | 1.57 | 189 | 0.8 | 193 | 2.19 | 189 | 1.02 | 7.6 | 219 | 15.84 | 210 | 214 | 1.02 | 1.04 | 0.98 |
| | 4 | C-4 | 1.13 | 136 | 0.0 | 136 | 0.00 | 131 | 1.03 | 0.0 | 136 | 0.00 | 131 | 120 | 1.13 | 1.03 | 1.09 |
| | | CF-4A | 1.1 | 132 | 0.8 | 137 | 3.27 | 134 | 1.02 | 7.6 | 155 | 17.06 | 148 | 155 | 1.00 | 1.05 | 0.95 |
| [24] | 5 | ST1 | 4.2 | 707 | 0.0 | 707 | 0.00 | 710 | 1.00 | 0.0 | 707 | 0.00 | 721 | 718 | 0.99 | 0.98 | 1.00 |
| | | ST2 | 4.2 | 712 | 0.1 | 714 | 0.26 | 724 | 0.99 | 3.7 | 780 | 9.58 | 745 | 740 | 1.05 | 1.05 | 1.01 |
| | | ST3 | 4.2 | 707 | 0.2 | 711 | 0.52 | 735 | 0.97 | 7.5 | 821 | 16.03 | 768 | 771 | 1.06 | 1.07 | 1.00 |
| | | ST4 | 4.2 | 707 | 0.3 | 713 | 0.79 | 743 | 0.96 | 11.3 | 841 | 18.83 | 803 | 782 | 1.07 | 1.05 | 1.03 |
| Average | | | | | | | | | 1.01 | | | | | | 1.05 | 1.03 | 1.02 |
| Standard deviation | | | | | | | | | 0.0227 | | | | | | 0.0532 | 0.0213 | 0.0535 |

Note: $t_s$ is the thickness of the steel tube, $N_{s,y}$ is the yield-bearing capacity for CHST short columns, $q_{f,y}$ is radial stress of the steel tube at the yield state, $N_{sf,y}$ is the theoretical value of the yield-bearing capacity of CFRP-CHST short columns, $W_{sf,y}$ is the improvement ratio of the yield load-bearing capacity, $N_{FE,y}$ is the finite element value of the yield load-bearing capacity of CFRP-CHST short columns, $q_{f,l}$ is radial stress of steel tube at the ultimate state, $N_{sf,l}$ is the theoretical value of the ultimate load-bearing capacity of CFRP-CHST short columns, $W_{sf,l}$ is the improvement ratio of the ultimate load-bearing capacity, $N_{FE,l}$ is the finite element value of the ultimate load-bearing capacity of CFRP-CHST short columns, $N_{e,l}$ is the experimental value of the ultimate load-bearing capacity of CFRP-CHST short columns.

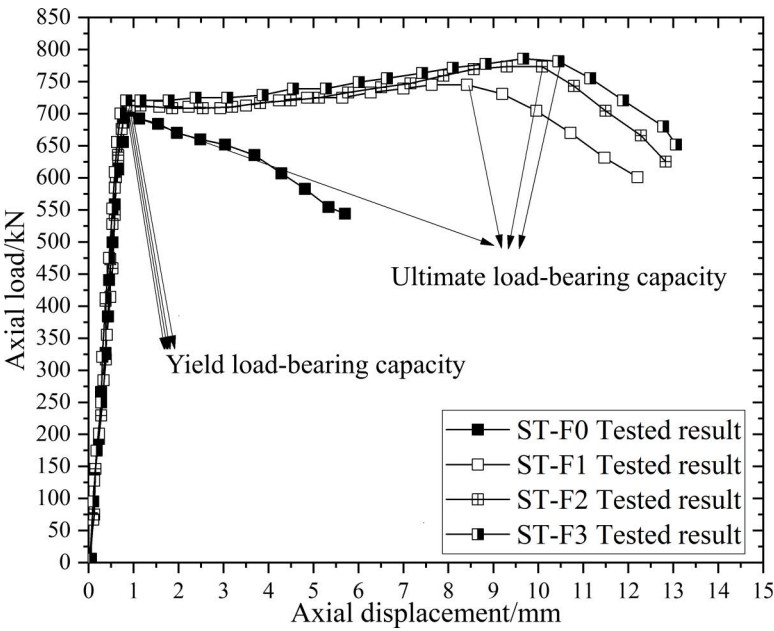

**Fig 7. Experimental results of the load-displacement curves from reference [24].**

to Eq. (20), the circumferential compressive stress $q_{f,l}$ provided by the CFRP to the steel tube at the ultimate strength state is several times greater than the compressive stress $q_{f,y}$ at the yield limit state (comparison results are shown in Table 3). This increased circumferential confinement significantly enhances the ultimate load-bearing capacity of the specimens.

By combining Eqs. (18) and (22), the ultimate load-bearing capacity $N_{s,y}$ of the unreinforced CHST short columns and the ultimate load-bearing capacity $N_{sf,l}$ of the CFRP-reinforced columns can be calculated. Consequently, the enhancement ratio of the ultimate load-bearing capacity due to CFRP reinforcement can be expressed as

$$W_{sf,l} = \frac{N_{sf,l} - N_{s,y}}{N_{s,y}} \times 100\% \tag{29}$$

The comparison results for the ultimate load-bearing capacity of CFRP-confined CHST short columns are presented in Table 3, including theoretical values $N_{sf,l}$, FE values $N_{FE,l}$, and experimental values $N_{e,l}$. The results reveal the following: the average ratio $N_{sf,l} / N_{e,l}$ is 1.05 with a standard deviation of 0.0532, the average ratio $N_{sf,l} / N_{FE,l}$ is 1.03 with a standard deviation of 0.0213, and the average ratio $N_{FE,l} / N_{e,l}$ is 1.02 with a standard deviation of 0.0535. All errors fall within a reasonable range, demonstrating that the proposed theoretical formula for ultimate load-bearing capacity is both accurate and reliable, and thus can be used as a reference for engineering design calculations.

The ultimate enhancement ratios for each specimen, calculated using Eq. (29), are presented in Table 3. It is observed that, within a certain range of parameters, the enhancement ratio of the ultimate load-bearing capacity of the specimens increases progressively with the number of CFRP layers. For instance, in the second group of specimens, the enhancement ratio for the 1H1L bonding configuration is 14.01%, whereas for the 2H2L configuration, it is 18.34%. The difference in enhancement ratios between these two configurations is 4.33% [19]. This pattern is consistent across other specimen groups and is corroborated by the experiments conducted by Teng and Hu [24].

From the load-displacement curves in Fig 7, it is evident that the ultimate load-bearing capacity generally increases with the number of GFRP layers among the four tested specimens. However, the increase in the number of GFRP layers results in diminishing improvements in the ultimate load-bearing capacity. Specifically, the specimen with three layers of GFRP wrapping (ST-F3) shows a significantly smaller increase in ultimate load-bearing capacity compared to specimens with one or two layers of GFRP wrapping (ST-F1 and ST-F2). This observation supports the theoretical conclusion that the enhancement in the ultimate load-bearing capacity of CHST short columns due to CFRP is limited. Despite this, CFRP wrapping technology holds significant potential for seismic retrofitting, as it enhances the structural resilience of CHST columns. The experimental results from Haedir and Zhao [19] generally indicate less ductility compared to those from Teng and Hu [24]. This discrepancy is attributed to Teng and Hu using GFRP with a lower modulus, which offers greater ultimate tensile strain and can endure larger deformations before fiber rupture.

## 5. Mechanism analysis

The previous analysis indicates that the enhancement of the ultimate load-bearing capacity of axially compressed CHST short columns due to CFRP reinforcement is limited. This limitation is quantified in the theoretical derivation by defining the CFRP confinement coefficient range, as detailed in Eq. (27). To understand the fundamental reasons behind this limitation, the following mechanism analysis integrates experimental results from references [19,24] with the finite element analysis results from this study.

The failure modes observed in the experiments by Haedir and Zhao [19] and in the finite element analysis conducted in this study reveal that, for identical CFRP wrapping configurations and number of layers (1H1L), a steel tube with greater thickness (CF-2A) exhibits a bulging failure mode (Fig 8a), while a steel tube with lesser thickness (CF-4A) shows inward buckling deformation away from the ends (Fig 8b). Similarly, Teng and Hu [24] report that for the same steel tube thickness, short columns with fewer layers of GFRP wrapping (SF-F1) typically fail through outward bulging (Fig 8c). As the number of GFRP layers increases (e.g., SF-F3), the failure mode transitions from outward bulging to inward buckling deformation away from the ends (Fig 8d).

The analysis indicates that the number of CFRP layers affects the failure modes of the specimens. The theoretical analysis reveals that an increased CFRP confining effect increasingly restrains the outward bulging deformation near the ends. Consequently, inward buckling deformation away from the ends becomes more pronounced. Since CFRP's primary advantage lies in its high tensile strength, its capacity to constrain inward buckling deformation is relatively limited. Therefore, when a component is primarily governed by inward buckling, additional CFRP layers do not significantly enhance the ultimate load-bearing capacity.

In summary, within a certain range, the ultimate load-bearing capacity of CFRP-reinforced CHST short columns shows a nonlinear increase with the number of CFRP layers, although this increase is relatively limited. When the CFRP confinement coefficient exceeds a certain threshold, adding additional CFRP layers does not significantly improve the ultimate load-bearing capacity of the CHST short columns. The theoretical analysis delineates the effective range for CFRP reinforcement of CHST short columns, offering valuable guidance for engineering design.

## 6. Parameter analysis

Six key parameters were selected for analysis: the thickness of the steel tube ($t_s$), the number of CFRP layers ($n_T$ and $n_L$), the yield strength of the steel tube ($f_y$), the tensile strength of the CFRP ($f_f$), the diameter-to-thickness ratio of the steel tube ($D/t_s$), and the CFRP

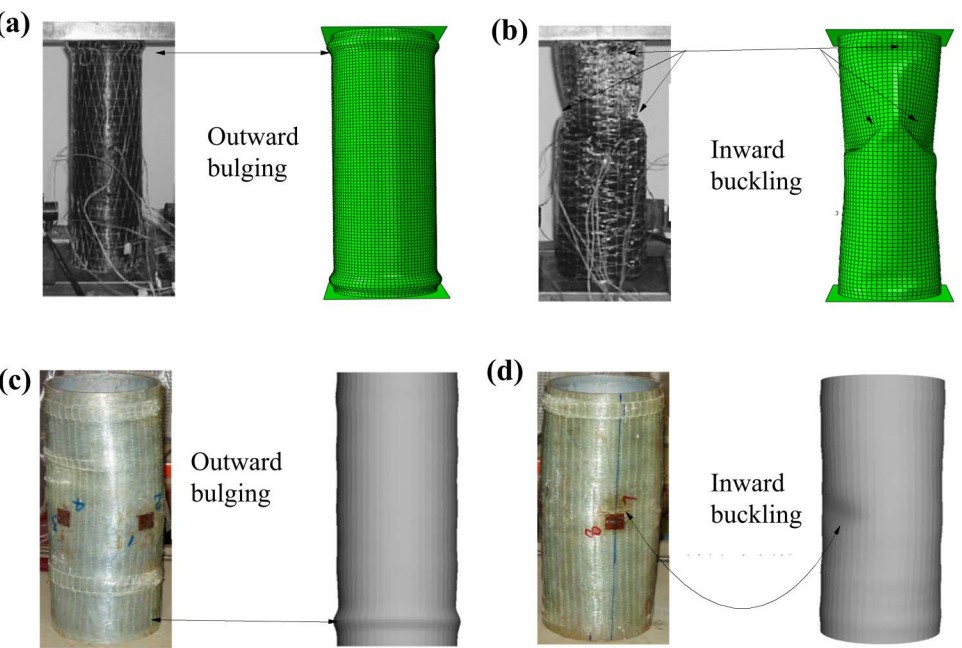

**Fig 8. Failure modes from experiments [19,24] and FE analysis: (a) CF-2A, (b) CF-4A, (c) SF-F1, and (d) SF-F3.**

confinement coefficient ( $\xi_f$ ). The rationale for selecting these parameters is as follows: $t_s$ directly affects local buckling and overall stability of the short column; $n_T$ and $n_L$ are the primary factors influencing the strengthening effect of the short column; $f_y$ and $f_f$ are critical for determining the structural response under axial loading; $D/t_s$ influences the load-bearing capacity and buckling behavior of the short column; and $\xi_f$ defines the effective range of CFRP strengthening. A total of 32 CFRP-CHST short columns with varying geometric and material parameters were analyzed using validated theoretical formulas. The specific model parameters are detailed in Table A1.

### 6.1. The thickness of the steel tube ( $t_s$ )

Fig 9a and 9b illustrate the variation in the yield and ultimate load-bearing capacities of CFRP-CHST short columns as the steel tube thickness changes. The data show that both the yield and ultimate load-bearing capacities increase with the steel tube thickness, demonstrating a clear linear relationship.

### 6.2. Yield strength of the steel tube ( $f_y$ )

Fig 10a and 10b illustrate the variation trends of the yield and ultimate load-bearing capacities of CFRP-CHST short columns under different steel tube yield strengths. The data points show a clear linear increase, indicating a positive correlation between the steel tube yield strength and both the yield and ultimate load-bearing capacities of the component.

### 6.3. Number of CFRP layers ( $n_T$ and $n_L$ )

Fig 11a shows the variation trend of the yield load-bearing capacity of CFRP-CHST short columns with different numbers of CFRP layers. The data points exhibit an approximately linear increase, suggesting a positive correlation between the number of CFRP layers and the yield

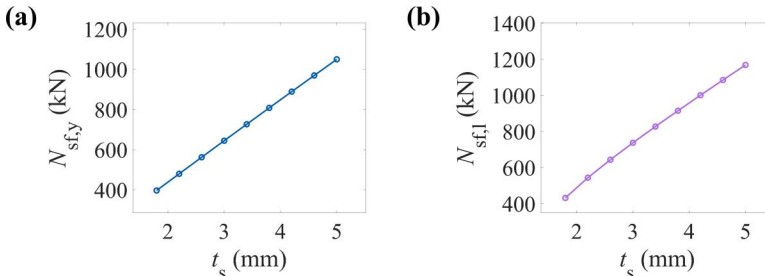

**Fig 9. Effect of the steel tube thickness on load-bearing capacity: (a) yield load-bearing capacity, and (b) ultimate load-bearing capacity.**

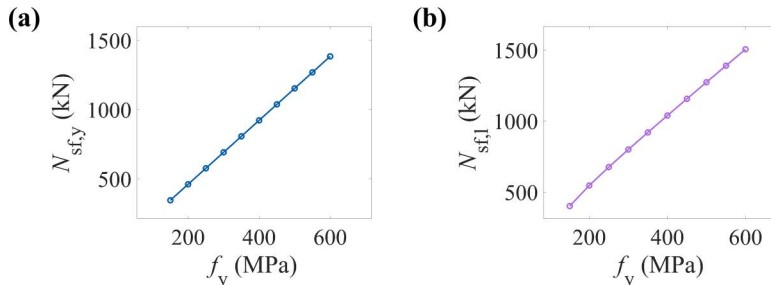

**Fig 10. Effect of steel yield strength on load-bearing capacity: (a) yield load-bearing capacity, and (b) ultimate load-bearing capacity.**

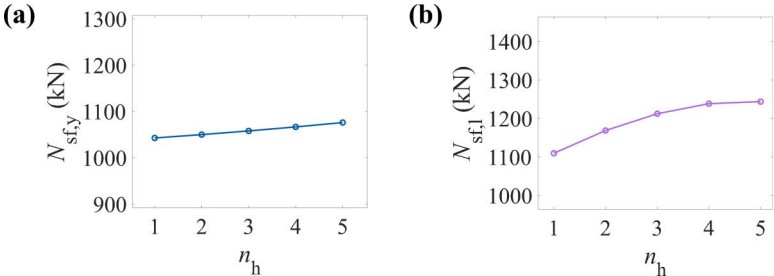

**Fig 11. Effect of the number of CFRP layers on load-bearing capacity: (a) yield load-bearing capacity, and (b) ultimate load-bearing capacity.**

load-bearing capacity. This trend likely results from the additional confinement provided by the CFRP layers, which enhances the component's stiffness and strength. Fig 11b illustrates the effect of CFRP layers on the ultimate load-bearing capacity of CFRP-CHST short columns. A nonlinear growth trend is observed, with the rate of increase in ultimate load-bearing capacity slowing as the number of CFRP layers rises. The data points stabilize after a certain number of layers, indicating a saturation point. Beyond this threshold, further increases in the number of CFRP layers have minimal impact on the ultimate load-bearing capacity. This suggests that the growth in ultimate load-bearing capacity reaches a limit, providing a basis for optimizing the number of CFRP layers and minimizing material waste.

### 6.4. Tensile strength of the CFRP ($f_f$)

Fig 12a shows the variation trend of the yield load-bearing capacity of CFRP-CHST short columns with different CFRP tensile strengths. The data points appear nearly horizontal, suggesting

that within the tested range of tensile strengths, the CFRP tensile strength has a negligible effect on the yield load-bearing capacity of the component. Fig 12b illustrates the impact of CFRP tensile strength on the ultimate load-bearing capacity of CFRP-CHST short columns. The data points display a clear nonlinear growth trend, indicating a positive correlation between the increase in CFRP tensile strength and the ultimate load-bearing capacity of the component.

## 6.5. The diameter-to-thickness ratio of the steel tube ($D/t_s$)

Fig 13a and 13b illustrate the variation trends of the yield and ultimate load-bearing capacities of CFRP-CHST short columns with different diameter-to-thickness ratios. The data points show a clear nonlinear decrease, indicating that as the diameter-to-thickness ratio increases, both the yield and ultimate load-bearing capacities of the component decrease significantly. At smaller diameter-to-thickness ratios, the load-bearing capacity decreases rapidly; however, as the ratio increases, the rate of decrease slows down. An increase in the steel tube diameter-to-thickness ratio may reduce the geometric stability of the component, leading to a decrease in its load-bearing capacity. When designing CFRP-CHST short columns, the diameter-to-thickness ratio should be carefully considered to ensure sufficient yield load-bearing capacity.

## 6.6. CFRP confinement coefficient ($\xi_f$)

Fig 14a shows the variation trend of the yield load-bearing capacity of CFRP-CHST short columns with different CFRP confinement coefficients. The data points display an approximately linear increase, indicating a positive correlation between the CFRP confinement coefficient and the yield load-bearing capacity of the component. The increase in the CFRP confinement coefficient enhances the yield load-bearing capacity of the CFRP-CHST short columns. In contrast, Fig 14b exhibits a nonlinear growth trend, suggesting that the rate of increase in ultimate

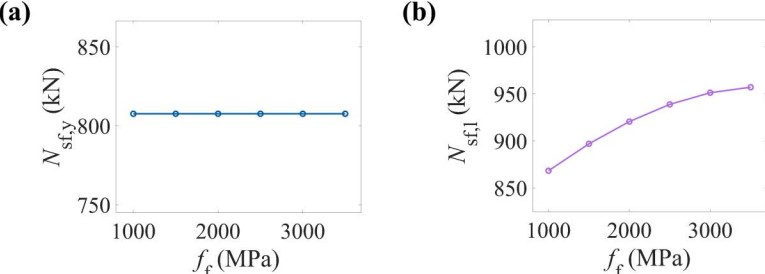

**Fig 12. Effect of CFRP tensile strength on load-bearing capacity: (a) yield load-bearing capacity, and (b) ultimate load-bearing capacity.**

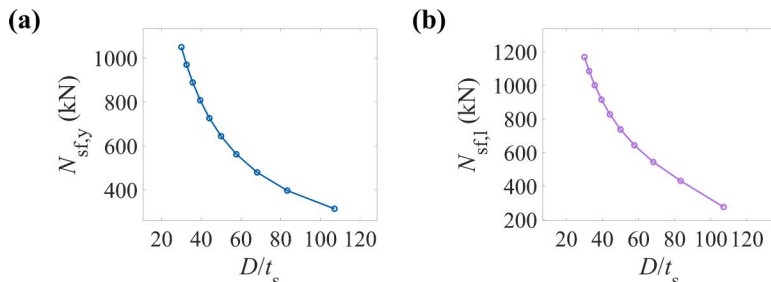

**Fig 13. Effect of the diameter-to-thickness ratio of the steel pipe on load-bearing capacity: (a) yield load-bearing capacity, and (b) ultimate load-bearing capacity.**

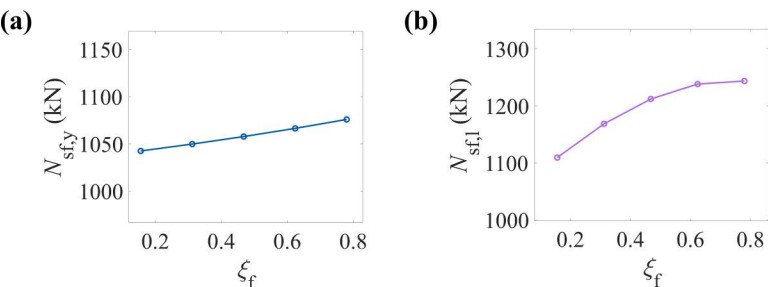

**Fig 14. Effect of CFRP confinement coefficient on load-bearing capacity: (a) yield load-bearing capacity, and (b) ultimate load-bearing capacity.**

load-bearing capacity slows as the CFRP confinement coefficient rises. The data points stabilize after reaching a certain confinement coefficient, potentially indicating a saturation point. Beyond this threshold, further increases in the CFRP confinement coefficient have a limited effect on enhancing the ultimate load-bearing capacity. This observation provides a basis for optimizing the CFRP confinement coefficient to avoid unnecessary material waste.

In the design of CFRP-CHST short columns, cost-effectiveness should be considered, and an appropriate number of CFRP layers and CFRP confinement coefficient selected to balance optimal load-bearing capacity and economic efficiency.

## 7. Conclusion

A comprehensive study has been conducted on the mechanical performance and failure mechanisms of CFRP-reinforced CHST short columns under axial compression, utilizing theoretical research, numerical analysis, and experimental data from the literature. The key findings are as follows:

(1) New theoretical formulas were developed to calculate the yield-bearing capacity (Eq. (18)) and ultimate bearing capacity (Eq. (22)) of CFRP-CHST short columns under axial compression, based on continuum mechanics and the limit equilibrium method. The validity of these formulas was confirmed through finite element simulations and experimental data. The average ratio of the calculated yield load to the simulation results was 1.01, with a standard deviation of 0.0227, and the ratio for the ultimate load was 1.03, with a standard deviation of 0.0210.

(2) Both theoretical and numerical analyses suggest that CFRP reinforcement has a limited effect on the yield-bearing capacity but significantly enhances the ultimate bearing capacity of CHST short columns, with some limitations. The yield load improvement due to CFRP reinforcement was minimal, with a maximum increase of only 3.86%. However, the enhancement in ultimate load was more pronounced, although the effect diminished as the number of CFRP layers increased. CFRP also improved the ductility of CHST short columns, which is advantageous for seismic strengthening. Thus, CFRP reinforcement shows substantial potential for seismic retrofitting applications.

(3) The failure modes of CFRP-CHST short columns were influenced by CFRP reinforcement, with the failure mode transitioning from localized outward bulging at the ends to inward buckling deformation as the number of CFRP layers increased. For specimens with identical CFRP configurations, thicker steel tubes exhibited localized end bulging, whereas thinner tubes showed inward buckling away from the ends. The failure mode shifted from outward bulging to inward buckling with an increase in CFRP layers.

(4) The CFRP confinement coefficient was introduced to quantify the enhancement of bearing capacity in CFRP-reinforced CHST short columns. A reference value for the optimal number of CFRP layers was proposed. When the CFRP confinement coefficient exceeds 1.155, further increases in the number of CFRP layers do not significantly improve the ultimate bearing capacity. In practical applications, the number of CFRP layers should be optimized based on the CFRP confinement coefficient to improve efficiency and reduce material waste. It is recommended that the CFRP confinement coefficient not exceed 1.155 to balance strengthening effectiveness and material costs.

Several areas require further exploration in future research:

(1) Further research is needed to investigate the mechanical behavior of CFRP-CHST short columns under bending and cyclic loading. This includes analyzing the impact of CFRP reinforcement on bending stiffness and flexural strength, as well as studying the hysteretic behavior and fatigue life under dynamic loading, such as during earthquakes. These studies would provide a comprehensive basis for seismic and wind resistance design.

(2) Future studies should expand to include CFRP-confined steel tube short columns with non-circular cross-sections (e.g., square, rectangular, and elliptical). Research could focus on developing theoretical models for these shapes, conducting experimental tests on their performance under various loading conditions, and validating the models through numerical simulations to explore stress-strain distributions and failure modes. This research would support the optimized design of such components in practical engineering, improving safety and cost-effectiveness.

(3) In-depth studies of CFRP-confined steel tube components with high slenderness ratios are needed. Research should aim to develop theoretical models for such short columns, conduct experimental tests to evaluate their performance under different loading conditions, and validate the models through numerical simulations. This research would enhance the safety and cost-effectiveness of slender and long-span structures.

## Nomenclature

| Symbol | Meaning | Unit |
|---|---|---|
| CFRP | Carbon fiber-reinforced polymer | – |
| GFRP | Glass fiber-reinforced polymer | – |
| CHST | Circular hollow steel tube | – |
| CFRP-CHST | CFRP-confined CHST short column | – |
| RHST | Rectangular hollow steel tube | – |
| SHST | Square hollow steel tube | – |
| CFST | Concrete-filled steel tube | – |
| FE | Finite element | – |
| $L$ | The length of the CFRP-CHST short column | mm |
| $\sigma_{zs}$ | Axial stress in the steel tube | MPa |
| $\sigma_{rs}$ | Radial stress in the steel tube | MPa |
| $\sigma_{,s}$ | Circumferential stress in the steel tube | MPa |
| $q_f$ | Circumferential confinement force provided by CFRP | MPa |
| $t_s$ | Thickness of the steel tube | mm |
| $d$ | The inner diameter of the steel tube | mm |
| $D$ | The outer diameter of the steel tube | mm |

| Symbol | Meaning | Unit |
|---|---|---|
| $f_y$ | Yield strength of steel | MPa |
| $E_{T,f}$ | Tensile modulus in the principal fiber direction | MPa |
| $E_{L,f}$ | Tensile modulus perpendicular to the principal fiber direction | MPa |
| $t_f$ | Thickness of the single-layer CFRP | mm |
| $n_T$ | Number of winding layers for circumferential fibres | – |
| $n_L$ | Number of winding layers for longitudinal fibers | – |
| $t_{eq,f}$ | Equivalent total thickness of the fibers | mm |
| $N$ | Bearing capacity of CHST | kN |
| $A_s$ | The cross-sectional area of the steel tube | mm$^2$. |
| $A_f$ | The cross-sectional area of fibres | mm$^2$ |
| $\varepsilon_{,s}$ | Circumferential strain of steel tube | με . |
| $E_s$ | Estic modulus of steel | MPa |
| $\mu_s$ | Poisson's ratio of steel | – |
| $\varepsilon_{,f}$ | Circumferential strain of fibers | με |
| $\sigma_{,f}$ | Circumferential stress of fibers | MPa |
| $q_{f,y}$ | Rial stress in the steel tube at yielding | MPa |
| $\psi$ , $\beta$ , $\gamma$ | Calculation parameters for radial stress | – |
| $\varepsilon_{zs}$ | Axial strain of steel tube | με |
| $\varepsilon_{zs,y}$ | Axial yield strain of the steel tube | με |
| $\Gamma$ | Calculation parameters for strain | – |
| $\varepsilon$ | Axial compressive strain | με |
| $\Delta L$ | The axial compression displacement of the component. | mm |
| $L$ | Total length of the component | mm |
| $\Delta l_{sf,y}$ | The theoretical value of the axial displacement at the yield point of the CFRP-CHST short column | mm |
| $\Delta l_{FE,y}$ | The finite element value of the axial displacement at the yield point of the CFRP-CHST short column | mm |
| $q_{f,y}$ | Radial stress of the steel tube at the yield state | MPa |
| $q_{f,l}$ | Radial stress of steel tube at the ultimate state | MPa |
| $f_f$ | The ultimate strength of the fibers | MPa |
| $\xi_f$ | CFRP confinement coefficient | – |
| $W_{sf,y}$ | Improvement ratio of the yield load-bearing capacity | % |
| $W_{sf,l}$ | Improvement ratio of the ultimate load-bearing capacity | % |
| $N_{s,y}$ | Yield bearing capacity for CHST short columns | kN |
| $N_{FE,y}$ | Finite element value of the yield load-bearing capacity of CFRP-CHST short columns | kN |
| $N_{sf,y}$ | The theoretical value of yield-bearing capacity of CFRP-CHST short columns | kN |
| $N_{e,l}$ . | The experimental value of the ultimate load-bearing capacity of CFRP-CHST short columns | kN |
| $N_{sf,l}$ | The theoretical value of the ultimate load-bearing capacity of CFRP-CHST short columns | kN |
| $N_{FE,l}$ | Finite element value of ultimate load-bearing capacity of CFRP-CHST short columns | kN |

**Table A1. Parameter analysis samples.**

| Specimen ID | Parameters of specimens | | | | | | | | | | | | Bearing capacity | |
|---|---|---|---|---|---|---|---|---|---|---|---|---|---|---|
| | $D$ /mm | $t_s$ /mm | $E_s$ /MPa | $\mu_s$ | $f_y$. /MPa | $E_{T,f}$ /MPa | $t_f$ /mm | $f_f$ /MPa | $D/t_s$ | $\xi_f$ | $n_H$. | $n_L$ | $N_{sf,y}$ /kN | $N_{sf,l}$ /kN |
| PA-1 | 150 | 1.4 | 209571 | 0.3 | 455 | 230000 | 0.176 | 2000 | 107 | 1.11 | 2 | 2 | 314 | 276 |
| PA-2 | 150 | 1.8 | 209571 | | | 230000 | 0.176 | 2000 | 83 | 0.87 | 2 | 2 | 397 | 432 |
| PA-3 | 150 | 2.2 | 209571 | | | 230000 | 0.176 | 2000 | 68 | 0.71 | 2 | 2 | 480 | 544 |
| PA-4 | 150 | 2.6 | 209571 | 0.3 | 455 | 230000 | 0.176 | 2000 | 58 | 0.60 | 2 | 2 | 562 | 644 |
| PA-5 | 150 | 3 | 209571 | 0.3 | 455 | 230000 | 0.176 | 2000 | 50 | 0.52 | 2 | 2 | 645 | 737 |
| PA-6 | 150 | 3.4 | 209571 | 0.3 | 455 | 230000 | 0.176 | 2000 | 44 | 0.46 | 2 | 2 | 727 | 827 |
| PA-7 | 150 | 3.8 | 209571 | 0.3 | 455 | 230000 | 0.176 | 2000 | 39 | 0.41 | 2 | 2 | 808 | 915 |
| PA-8 | 150 | 4.2 | 209571 | 0.3 | 455 | 230000 | 0.176 | 2000 | 36 | 0.37 | 2 | 2 | 889 | 1001 |
| PA-9 | 150 | 4.6 | 209571 | 0.3 | 455 | 230000 | 0.176 | 2000 | 33 | 0.34 | 2 | 2 | 970 | 1085 |
| PA-10 | 150 | 5 | 209571 | 0.3 | 455 | 230000 | 0.176 | 2000 | 30 | 0.31 | 2 | 2 | 1050 | 1169 |
| PA-11 | 150 | 5 | 209571 | 0.3 | 455 | 230000 | 0.176 | 2000 | 30 | 0.16 | 1 | 1 | 1043 | 1109 |
| PA-12 | 150 | 5 | 209571 | 0.3 | 455 | 230000 | 0.176 | 2000 | 30 | 0.31 | 2 | 2 | 1050 | 1169 |
| PA-13 | 150 | 5 | 209571 | 0.3 | 455 | 230000 | 0.176 | 2000 | 30 | 0.47 | 3 | 3 | 1058 | 1212 |
| PA-14 | 150 | 5 | 209571 | 0.3 | 455 | 230000 | 0.176 | 2000 | 30 | 0.62 | 4 | 4 | 1066 | 1238 |
| PA-15 | 150 | 5 | 209571 | 0.3 | 455 | 230000 | 0.176 | 2000 | 30 | 0.78 | 5 | 5 | 1076 | 1244 |
| PA-16 | 150 | 5 | 209571 | 0.3 | 455 | 230000 | 0.176 | 2000 | 30 | 0.94 | 6 | 6 | 1086 | 1222 |
| PA-17 | 150 | 5 | 209571 | 0.3 | 150 | 230000 | 0.176 | 2000 | 30 | 0.95 | 2 | 2 | 346 | 402 |
| PA-18 | 150 | 5 | 209571 | 0.3 | 200 | 230000 | 0.176 | 2000 | 30 | 0.71 | 2 | 2 | 461 | 547 |
| PA-19 | 150 | 5 | 209571 | 0.3 | 250 | 230000 | 0.176 | 2000 | 30 | 0.57 | 2 | 2 | 577 | 676 |
| PA-20 | 150 | 5 | 209571 | 0.3 | 300 | 230000 | 0.176 | 2000 | 30 | 0.47 | 2 | 2 | 692 | 800 |
| PA-21 | 150 | 5 | 209571 | 0.3 | 350 | 230000 | 0.176 | 2000 | 30 | 0.41 | 2 | 2 | 808 | 921 |
| PA-22 | 150 | 5 | 209571 | 0.3 | 400 | 230000 | 0.176 | 2000 | 30 | 0.35 | 2 | 2 | 923 | 1039 |
| PA-23 | 150 | 5 | 209571 | 0.3 | 450 | 230000 | 0.176 | 2000 | 30 | 0.32 | 2 | 2 | 1038 | 1157 |
| PA-24 | 150 | 5 | 209571 | 0.3 | 500 | 230000 | 0.176 | 2000 | 30 | 0.28 | 2 | 2 | 1154 | 1274 |
| PA-25 | 150 | 5 | 209571 | 0.3 | 550 | 230000 | 0.176 | 2000 | 30 | 0.26 | 2 | 2 | 1269 | 1390 |
| PA-26 | 150 | 5 | 209571 | 0.3 | 600 | 230000 | 0.176 | 2000 | 30 | 0.24 | 2 | 2 | 1384 | 1506 |
| PA-27 | 150 | 5 | 209571 | 0.3 | 350 | 230000 | 0.176 | 1000 | 30 | 0.20 | 2 | 2 | 808 | 868 |
| PA-28 | 150 | 5 | 209571 | 0.3 | 350 | 230000 | 0.176 | 1500 | 30 | 0.30 | 2 | 2 | 808 | 897 |
| PA-29 | 150 | 5 | 209571 | 0.3 | 350 | 230000 | 0.176 | 2000 | 30 | 0.41 | 2 | 2 | 808 | 921 |
| PA-30 | 150 | 5 | 209571 | 0.3 | 350 | 230000 | 0.176 | 2500 | 30 | 0.51 | 2 | 2 | 808 | 939 |
| PA-31 | 150 | 5 | 209571 | 0.3 | 350 | 230000 | 0.176 | 3000 | 30 | 0.61 | 2 | 2 | 808 | 951 |
| PA-32 | 150 | 5 | 209571 | 0.3 | 350 | 230000 | 0.176 | 3500 | 30 | 0.71 | 2 | 2 | 808 | 957 |

Note: $D$ is the outer diameter of the steel tube, $t_s$ is the thickness of the steel tube, $E_s$ is the elastic modulus of steel, $\mu_s$ is the Poisson's ratio of steel, $f_y$ is the yield strength of steel, $E_{T,f}$ is tensile modulus in the principal fiber direction, $t_f$ is the thickness of the single-layer CFRP, $f_f$ is the ultimate strength of the fibers, $\xi_f$ is the CFRP confinement coefficient, $n_H$ is the number of winding layers for circumferential fibres, $n_L$ is the number of winding layers for longitudinal fibers, $N_{sf,y}$ is the theoretical value of the yield-bearing capacity of CFRP-CHST short columns, $N_{sf,l}$ is the theoretical value of the ultimate load-bearing capacity of CFRP-CHST short columns.

# Appendix A

## Parameter analysis samples

A total of 32 CFRP-CHST short-column specimens with different geometric and material parameters were analyzed through the validated theoretical formulas. The specific model parameters are provided in Table A1.

## Supporting information

**S1 File. Original data.**
(XLSX)

## Author contributions

**Conceptualization:** Jian Chen, Hairong Huang.

**Data curation:** Jian Chen, Hairong Huang, Yun Zhou, Kan Liu.

**Formal analysis:** Jian Chen, Hairong Huang, Yun Zhou, Kan Liu.

**Funding acquisition:** Jian Chen, Hairong Huang.

**Investigation:** Jian Chen.

**Methodology:** Jian Chen, Yun Zhou, Kan Liu.

**Project administration:** Jian Chen, Hairong Huang.

**Resources:** Jian Chen, Hairong Huang, Yun Zhou.

**Software:** Jian Chen, Hairong Huang, Yun Zhou, Kan Liu.

**Validation:** Jian Chen, Hairong Huang, Kan Liu.

**Visualization:** Jian Chen, Hairong Huang, Kan Liu.

**Writing – original draft:** Jian Chen, Yun Zhou.

**Writing – review & editing:** Hairong Huang.

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
