## [Decision Letter · Decision Letter 0]

4 Dec 2024

PONE-D-24-42244Axial compression behavior and failure mechanism of CFRP-confined circular hollow steel tube short columns: theoretical and numerical analysisPLOS ONE

Dear Dr. Huang,

Thank you for submitting your manuscript to PLOS ONE. After careful consideration, we feel that it has merit but does not fully meet PLOS ONE’s publication criteria as it currently stands. Therefore, we invite you to submit a revised version of the manuscript that addresses the points raised during the review process.

We look forward to receiving your revised manuscript.

Kind regards,

Mohammadreza Vafaei, Ph.D.

Academic Editor

PLOS ONE

Journal Requirements:

2. Thank you for stating the following financial disclosure: “This work was supported by the General Research Project of Zhejiang Provincial Department of Education (No. Y202352472)”

3. Thank you for stating the following in the Competing Interests section: “On behalf of all authors, disclose any competing interests that could be perceived to bias this work—acknowledging all financial support and any other relevant financial or non-financial competing interests.”

Please confirm that this does not alter your adherence to all PLOS ONE policies on sharing data and materials, by including the following statement: "This does not alter our adherence to PLOS ONE policies on sharing data and materials.” (as detailed online in our guide for authors http://journals.plos.org/plosone/s/competing-interests). If there are restrictions on sharing of data and/or materials, please state these. Please note that we cannot proceed with consideration of your article until this information has been declared. Please include your updated Competing Interests statement in your cover letter; we will change the online submission form on your behalf.

4. We note that your Data Availability Statement is currently as follows: “All relevant data are within the manuscript and in Supporting Information files.”

Please confirm at this time whether or not your submission contains all raw data required to replicate the results of your study. Authors must share the “minimal data set” for their submission. PLOS defines the minimal data set to consist of the data required to replicate all study findings reported in the article, as well as related metadata and methods (https://journals.plos.org/plosone/s/data-availability#loc-minimal-data-set-definition). For example, authors should submit the following data: - The values behind the means, standard deviations and other measures reported; - The values used to build graphs; - The points extracted from images for analysis. Authors do not need to submit their entire data set if only a portion of the data was used in the reported study. If your submission does not contain these data, please either upload them as Supporting Information files or deposit them to a stable, public repository and provide us with the relevant URLs, DOIs, or accession numbers. For a list of recommended repositories, please see https://journals.plos.org/plosone/s/recommended-repositories. If there are ethical or legal restrictions on sharing a de-identified data set, please explain them in detail (e.g., data contain potentially sensitive information, data are owned by a third-party organization, etc.) and who has imposed them (e.g., an ethics committee). Please also provide contact information for a data access committee, ethics committee, or other institutional body to which data requests may be sent. If data are owned by a third party, please indicate how others may request data access.

6. Please note that funding information should not appear in any section or other areas of your manuscript. We will only publish funding information present in the Funding Statement section of the online submission form. Please remove any funding-related text from the manuscript.

Additional Editor Comments:

Address all reviewer comments, particularly emphasizing the novelty and contribution of your work.

Reviewers' comments:

Reviewer's Responses to Questions

**Comments to the Author**

1. Is the manuscript technically sound, and do the data support the conclusions?

Reviewer #1: Partly

Reviewer #2: Partly

Reviewer #3: Partly

2. Has the statistical analysis been performed appropriately and rigorously? 

Reviewer #1: No

Reviewer #2: I Don't Know

Reviewer #3: Yes

3. Have the authors made all data underlying the findings in their manuscript fully available?

Reviewer #1: No

Reviewer #2: Yes

Reviewer #3: No

4. Is the manuscript presented in an intelligible fashion and written in standard English?

Reviewer #1: Yes

Reviewer #2: No

Reviewer #3: Yes

5. Review Comments to the Author

Reviewer #1: This paper presents a study that combines theoretical research with numerical analysis to evaluate the effectiveness of CFRP reinforcement in CHST short columns under axial compression. New formulas for calculating the yield and ultimate bearing capacities of CFRP-CHST short columns are proposed and validated through comparisons with experimental results and finite element analysis. It is essential to highlight the main contributions of this research while clearly differentiating the work conducted by others from the new insights presented in this study. The following issues need to be addressed for clarification.

- Acronyms and abbreviations need to be defined when they are mentioned for the first time. Acronyms are used in many locations without definition. Symbols are used without definition. Please add a list for the symbols used.

- Please include an illustration for cross-sectional information (D, d, ts, and L) in Fig 1.

- On page 12, the authors summarized the specimens tested by Haedir and Zhao [18], but they did not provide any information about the specimens used in the experiments conducted by Teng and Hu [23]. A Comprehensive Table Considering All Specimens Should Be Included. Additionally, Tables 4 and 5 do not include comparisons of the specimens tested by Teng and Hu [23].

- It is imperative to include the specified parameters and to provide a clear rationale for their selection, along with a detailed explanation of their anticipated impact on the behavior of circular CFRP-CHST short columns. This will significantly enhance understanding. The parametric study must encompass a comprehensive range of parameters to thoroughly examine their effects on theoretical formula of the yield and ultimate bearing capacities of CFRP-CHST short columns.

- How interaction between the steel and CFRP was modelled?

- Line 240-241, “the two loading plates at the ends are modeled as analytical rigid bodies”. Please explain in the manuscript.

- The study only considers CFRP-CHST short columns under axial compression, not addressing other loading conditions such as flexure or cyclic loading. Mention the limitations regarding loading conditions explicitly and propose future studies to include other loading conditions. Highlight the potential impact of these conditions on the behavior of CFRP-CHST short columns.

- The study is focused on circular CFRP-CHST columns, limiting the generalizability of the conclusions to non-circular cross-sections. Acknowledge this limitation and suggest extending the study to non-circular cross-sections. Discuss how the two formulas might differ for other geometries and propose a methodology for investigating these cases.

- More discussion about the effect of CFRP effect with higher slenderness ratios is required.

- The references cited are largely irrelevant to the research topic, especially references from [1] to [10]. It is crucial that these references be updated to include more recent and pertinent sources that directly relate to the subject matter.

Reviewer #2: From my perspective, this paper is highly repetitive and lacks novelty. The results presented are merely reiterations of existing findings that have been previously collected, analyzed, and modeled by our team through both numerical and machine learning approaches. Unfortunately, there is no significant new contribution in this study. Extensive literature already covers similar numerical and theoretical work on these types of specimens, and existing models adequately address stress-strain relationships for these materials. Given this context, I do not believe this paper meets the standards for publication in any scholarly journal.

Reviewer #3: Title of Paper: Axial compression behavior and failure mechanism of CFRP-confined circular hollow steel tube short columns

Manuscript Number: PONE-D-24-42244

This manuscript has taken CFRP-CHST short columns into consideration and studied the structural behaviour of such columns under axial compression. The study is well organized and the authors tried to investigate the concept of their study through different methods. While the novelty of the study is a bit poor, the package of the whole study is considerable from my point of view. However, it needs some amendments to fulfill the requirements of publishing in the journal.

Comments

1. Lines 106-109: This part needs to be polished in terms of language.

2. Fig. 2: The stress distribution is shown separately for the steel tube and the CFRP. Since I have seen this type of illustration a lot, I suggest showing the stress distribution on a section of CFRP-CHST. Can you do it?

3. For the equations you need to clarify each parameter beneath the formula.

4. Equations 5 and 18: It seems the authors just considered the area of steel. What about the impact of carbon fiber reinforced polymer? Has been ignored?!

5. Line 136: What is assumption 2? Please explain.

6. Lines 124, 126, 168: The terms of bearing capacity, yield bearing capacity, ultimate bearing capacity were used and the final equations 5, 18, 22 are a bit confusing! You can provide and highlight a final formula as the outcome of this study. I think would be user-friendly.

7. Equations 5, 18, 22: Still, I could not recognize the area of CFRP in these equations. Please clarify.

8. Table 1: What are nl , nh ?

9. Table 2: I think it is unnecessary as the necessary details of CFRP are already mentioned in Section 3.1.

10. Line 253 needs a space.

11. Fig 4: needs some adjustments. One arrow is missing and b) overlaps with the column.

12. Table 3: The information here can be written within the manuscript and there is no need for a single-row Table here!

13. I think there is no need for 3 sub-sections for Section. 3 as all of the subsections are very short.

14. Line 292: Do you mean Yield?

15. Line 301: “Comparison of longitudinal displacement at yield for CHST short columns”, at yield level/point.

16. Table 5: It needs a footnote explaining the parameters.

17. Conclusion Line 6: “A theoretical formula “, please highlight which one.

18. Conclusion No. 3: How this is concluded? From FEA?

19. I suggest to re-write the conclusions with specific details. The conclusion looks broad and not only for this study.

20. Line 25: In the submission process, it is mentioned that the data is available within the manuscript and supplementary information, and the statement “Data will be made available on request” contradicts this.

21. The references are very limited and almost 10 of 23 are outdated. It is usually suggested to reference articles older than ten years for only 10 percent of the total citations.

6. PLOS authors have the option to publish the peer review history of their article (what does this mean? ). If published, this will include your full peer review and any attached files.

**Do you want your identity to be public for this peer review?** For information about this choice, including consent withdrawal, please see our Privacy Policy .

Reviewer #1: No

Reviewer #2: No

Reviewer #3: **Yes: ** Payam Sarir

---

## [Author Response · Author response to Decision Letter 0]

19 Jan 2025

Dear Reviewers,

I would like to express sincere gratitude for your insightful and constructive feedback on our manuscript titled "Axial compression behavior and failure mechanism of CFRP-confined circular hollow steel tube short columns: theoretical and numerical analysis." Your comprehensive and thoughtful comments have significantly enhanced the depth and scientific rigor of our research. We appreciate your dedication to the peer review process. All your suggestions and concerns have been thoroughly reviewed and addressed. The revised manuscript incorporates these suggestions, with the modified content highlighted in blue for easy reference.

Additionally, a detailed point-by-point response to each reviewer's comment has been prepared. Reviewer comments are presented in italics, followed by corresponding responses. Given the extensive content, the authors' responses to all constructive feedback are provided in the attached document titled "Response to Reviewers".

We kindly request your esteemed review of the revised manuscript at your earliest convenience. We sincerely hope that these changes meet your expectations for publication, and we await your final evaluation of our revised manuscript.

Best regards,

Jian Chen, Hairong Huang, Yun Zhou, Kan Liu

---

## [Decision Letter · Decision Letter 1]

13 Feb 2025

Axial compression behavior and failure mechanism of CFRP-confined circular hollow steel tube short columns: theoretical and numerical analysis

PONE-D-24-42244R1

Dear Dr. Hairong Huang,

We’re pleased to inform you that your manuscript has been judged scientifically suitable for publication and will be formally accepted for publication once it meets all outstanding technical requirements.

Kind regards,

Mohammadreza Vafaei, Ph.D.

Academic Editor

PLOS ONE

Additional Editor Comments (optional):

Reviewers' comments:

Reviewer's Responses to Questions

**Comments to the Author**

1. If the authors have adequately addressed your comments raised in a previous round of review and you feel that this manuscript is now acceptable for publication, you may indicate that here to bypass the “Comments to the Author” section, enter your conflict of interest statement in the “Confidential to Editor” section, and submit your "Accept" recommendation.

Reviewer #1: All comments have been addressed

Reviewer #3: All comments have been addressed

2. Is the manuscript technically sound, and do the data support the conclusions?

Reviewer #1: Yes

Reviewer #3: Yes

3. Has the statistical analysis been performed appropriately and rigorously? 

Reviewer #1: Yes

Reviewer #3: Yes

4. Have the authors made all data underlying the findings in their manuscript fully available?

Reviewer #1: Yes

Reviewer #3: Yes

5. Is the manuscript presented in an intelligible fashion and written in standard English?

Reviewer #1: Yes

Reviewer #3: Yes

6. Review Comments to the Author

Reviewer #1: The authors have responded well. The revised manuscript can be considered for an acceptance in journal

Reviewer #3: Dear authors,

Thank you for addressing all the comments and consider the details in revising the manuscript.

All the best.

7. PLOS authors have the option to publish the peer review history of their article (what does this mean? ). If published, this will include your full peer review and any attached files.

**Do you want your identity to be public for this peer review?** For information about this choice, including consent withdrawal, please see our Privacy Policy .

Reviewer #1: **Yes: ** Asmaa Y Hamed

Reviewer #3: **Yes: ** Payam Sarir

---

## [Editor Report · Acceptance letter]

PONE-D-24-42244R1

PLOS ONE

Dear Dr. Huang,

I'm pleased to inform you that your manuscript has been deemed suitable for publication in PLOS ONE. Congratulations! Your manuscript is now being handed over to our production team.

Kind regards,

on behalf of

Dr. Mohammadreza Vafaei

Academic Editor

PLOS ONE